# Sentinel Lymph Node Detection in Cutaneous Melanoma Using Indocyanine Green-Based Near-Infrared Fluorescence Imaging: A Systematic Review and Meta-Analysis

**DOI:** 10.3390/cancers16142523

**Published:** 2024-07-12

**Authors:** Marcus Wölffer, Rémy Liechti, Mihai Constantinescu, Ioana Lese, Cédric Zubler

**Affiliations:** 1Department of Orthopaedics, Hand and Trauma Surgery, Stadtspital Waid, Tièchestrasse 99, 8037 Zürich, Switzerland; 2Department of Plastic and Hand Surgery, University of Bern, Inselspital University Hospital Bern, Freiburgstrasse 18, 3010 Bern, Switzerland; remy.liechti@insel.ch (R.L.); mihai.constantinescu@insel.ch (M.C.); ioana.lese@insel.ch (I.L.); cedric.zubler@insel.ch (C.Z.)

**Keywords:** melanoma, sentinel lymph node biopsy, indocyanine green, near-infrared fluorescence imaging, ICG, meta-analysis, dermato-oncology

## Abstract

**Simple Summary:**

A positive sentinel lymph node biopsy of cutaneous melanoma patients has a substantial impact on subsequent treatment decisions. The standard of care approach to identify the sentinel lymph node is technetium (Tc)-based lymphoscintigraphy. This technique comes with a radiation exposure and high costs. Indocyanine green (ICG)-based near-infrared fluorescence imaging could be an alternative if demonstrated to have a comparable diagnostic accuracy. Therefore, a systematic literature review and meta-analysis were conducted considering studies comparing the accuracy of ICG and Tc for intraoperative guidance. Within the seven included studies, no significant differences between the two modalities were found regarding the identification of metastatic patients or the false negative rate. ICG may be a non-inferior alternative to Tc for intraoperative identification of the sentinel lymph node in cutaneous melanoma patients.

**Abstract:**

The standard of care approach to identify sentinel lymph nodes (SLNs) in clinically non-metastatic cutaneous melanoma patients is technetium (Tc)-based lymphoscintigraphy. This technique is associated with radiation exposure, a long intervention time, high costs, and limited availability. Indocyanine green (ICG)-based near-infrared fluorescence imaging offers a potential alternative if proven to be of comparable diagnostic accuracy. While several clinical cohorts have compared these modalities, no systematic review exists that provides a quantitative analysis of their results. Hence, a systematic literature review was conducted in December 2023 considering clinical studies comparing the diagnostic accuracy of ICG and Tc for sentinel lymph node biopsy in cutaneous melanoma patients. Three hundred nineteen studies were identified and further screened in accordance with the PRISMA 2020 guidelines, resulting in seven studies being included in the final meta-analysis. Tc identified a significantly higher number of SLNs and metastatic SLNs in prospective studies only. However, in the overall meta-analysis of all included comparative studies, no significant differences were found regarding the identification of metastatic patients or the false negative rate (FNR). ICG may be a non-inferior alternative to Tc for intraoperative guidance in sentinel lymph node biopsy in cutaneous melanoma patients. Future randomized controlled trials are needed, especially regarding the preoperative, transcutaneous identification of the affected lymph node basin.

## 1. Introduction

Cutaneous melanoma is one of the most common malignancies after breast, prostate, lung, and colon cancers [1]. A sentinel lymph node biopsy (SLNB) is a crucial step in the management of cutaneous melanoma patients with clinically localized disease, as it provides information about the oncologic prognosis. It allows for adequate staging and indicates adjuvant therapy with immune checkpoint inhibitors or targeted therapy in the case of a metastatic sentinel lymph node (SLN) [2].

The standard of care approach to identify sentinel lymph nodes in clinically non-metastatic cutaneous melanoma patients is lymphoscintigraphy (LS), following the injection of a technetium (Tc) tracer [3]. However, this technique is associated with certain disadvantages. Cost analyses including the expenses for travel, contrast medium, medical staff, and hospital infrastructure have demonstrated the poor cost-effectiveness of this technique [4]. Additionally, even if low, there is a radioactive burden for patients and medical personnel alike [5]. SLNB is also hampered by the limited availability of Tc-based LS in developing countries [6], stressing the need for alternative procedures.

Indocyanine green (ICG)-based near-infrared fluorescence imaging, nowadays mainly used in conjunction with Tc, offers a potential alternative. In other fields of oncologic surgery, this method has already been investigated and proven safe and effective. Indeed, several non-inferiority trials in breast cancer research have revealed ICG to have an SLN detection rate at least equivalent to Tc [7,8]. A recent meta-analysis even concluded that ICG alone is better than blue dye or Tc alone, and not worse than blue dye and Tc combined in SLNBs for patients with early-stage breast cancer [9], yet this has not been conclusively investigated for cutaneous melanoma.

If proven to be safe and of comparable diagnostic accuracy, ICG would allow for a significantly cheaper, faster, and radiation-free patient pathway. While a number of clinical cohorts have reported on the direct comparison between the ICG and Tc techniques, to the best of our knowledge, no systematic review exists that provides a quantitative analysis of these results. This systematic review and meta-analysis aim to investigate the respective diagnostic accuracy of ICG-only and Tc-only approaches in identifying cutaneous melanoma SLNs, metastatic SLNs, and metastatic patients, with special regard to the false-negative rate (FNR), which is of utmost significance to affected patients.

## 2. Materials and Methods

This systematic review and meta-analysis was conducted according to the Preferred Reporting Items for Systematic Reviews and Meta-analysis (PRISMA) and the Meta-analysis of Observational Studies in Epidemiology (MOOSE) checklists [10,11]. Due to the nature of this investigation, no ethical approval was required. The study was not registered.

### 2.1. Search Strategy and Selection Criteria

Data collection was performed according to the principles laid out by the Cochrane Collaboration [12]. Keyword selection was based on the PICO model [13]. In December 2023, MEDLINE via PubMed (National Library of Medicine, Bethesda, MD, USA), Embase, and the Cochrane Library databases were systematically searched for studies comparing the diagnostic accuracy of Tc-based LS and ICG-based near-infrared fluorescence imaging in identifying SLNs in cutaneous melanoma patients. The search terms used described the study population and intervention: ((melanom*) OR (malignant lentigo) OR (lentigo maligna)) AND ((icg) OR (indocyanine green) OR (Near-Infrared fluorescence) OR (near infrared fluorescence) OR (NIRFI)) AND ((sentinel) OR (lymph*) OR (SLN)). The exclusion criteria were: language other than English, missing full-text publication, no original outcome on the diagnostic accuracy of ICG and Tc in cutaneous melanoma patients, wrong study type (reviews, study protocols, case reports, published abstracts from scientific meetings, letters, and animal studies), inclusion of fewer than 10 cases, malignancies other than melanoma, use of a hybrid ICG–Tc tracer, and concomitant use of dyes other than ICG. Two reviewers (MW and CZ) independently performed full-text screening. Any disagreements on the eligibility of full-text articles were resolved by a senior author (IL). An extensive cross-check of the references from the original studies was performed to identify potential additional articles.

### 2.2. Data Extraction

Two reviewers (MW, CZ) independently performed the data extraction. The following baseline characteristics were extracted from the included studies: first author, year of publication, study design, type of diagnostic intervention, prospective or retrospective nature of the study, multi- or monocenter study, study period in months, number of patients, mean age in years, number of men and women, primary tumor location, moment of SLN identification with ICG (transcutaneous or after incision), excision of SLNs based on each method, camera system used, ICG dose used, number of metastatic patients with respective number identified by: both methods, ICG only, Tc only, ICG in total, Tc in total; number of SLNs sampled with respective number identified by: both methods, ICG only, Tc only, ICG in total, Tc in total; number of metastatic SLNs with respective number identified by: both methods, ICG only, Tc only, ICG in total, Tc in total; mean, median, minimum and maximum follow-up with corresponding standard deviation in months; and number of patients with recurrences in previously sampled, negative SLN basin.

### 2.3. Quality Assessment

Two reviewers (MW, CZ) assessed the methodological quality of the seven included studies using the Methodological Index for Non-Randomized Studies (MINORS) independently [14]. Disagreements were resolved by consensus with RL and IL.

### 2.4. Study Outcomes

The primary outcome of interest was the FNR of ICG and Tc, respectively. We defined a false negative event as a recurrence in a previously sampled, negative SLN basin during follow-up or a positive, i.e., metastatic, SLN identified only by the other method.

The FNR for ICG or Tc was calculated accordingly: (number of patients with a recurrence in a previously sampled, negative SLN basin + number of patients with a positive, i.e., metastatic, SLN identified only by the other method)/(number of patients with a recurrence in a previously sampled, negative SLN basin + number of patients with a positive, i.e., metastatic, SLN identified only by the other method + number of true positive patients with the respective method). In addition to calculating the FNRs of ICG and Tc for each study, a meta-FNR for ICG and Tc was calculated.

Five of the seven included studies provided a data set sufficient for calculating the FNRs of ICG and Tc. A mean or median follow-up period of at least 12 months was deemed necessary for a realistic assessment of the FNR. Previous multicentre studies have shown that a significant number of recurrences after initially negative sentinel lymph node biopsies are diagnosed within this timeframe [15]. Thus, four studies were finally included in the FNR meta-analysis.

Secondary outcomes were the total number of SLNs identified by ICG and Tc, the number of metastatic SLNs identified by ICG and Tc, and the number of metastatic patients identified by ICG and Tc. In that respect, all the included studies provided sufficient data to qualify for quantitative meta-analysis.

### 2.5. Statistical Analysis

Information about continuous variables was presented as the means with standard deviation, or the information was converted to the mean and standard deviation following the methods described in the Cochrane Handbook for Systematic Reviews of Interventions [12]. Weighted means were calculated for the synthesis of continuous variables to account for differences in study sizes. Dichotomous variables were presented as counts and percentages. Relative risk and number needed to harm calculations and the corresponding 95% confidence interval for the primary outcome (FNR) were performed according to Altman 1998 [16]. Effects of tracing options (Tc or ICG) on continuous outcomes were pooled using the random effects inverse variance weighting method and presented as the mean difference with a corresponding 95% confidence interval. The effects of the tracing options on binary outcomes were pooled using the random effects Mantel–Haenszel method and presented as an odds ratio with a corresponding 95% confidence interval. A *p*-value below 0.05 was considered statistically significant. Heterogeneity between studies was assessed by visual inspection of forest plots (overlapping of 95% confidence interval) and by the I^2^ statistic for heterogeneity. Review Manager (RevMan, version 5.3.5) was used for all the statistical analyses.

## 3. Results

### 3.1. Literature Search

A total of 319 publications were initially identified and imported into Rayyan [17] for further selection. Then, 105 duplicate records were removed before the primary review. The remaining 214 records were subjected to a title and abstract screening by two reviewers independently (MW, CZ). Of these, 191 records were excluded, and 23 full-text reports were successfully retrieved. During the secondary review, 16 reports were excluded due to the additional intraoperative use of blue dye, imprecise separation of melanoma or ICG subgroups, missing data despite contacting the corresponding author, and studies reporting on the same patient cohort. Thus, after screening and exclusion in accordance with the PRISMA 2020 guidelines [10], a total of seven studies remained. The search syntax is demonstrated in Figure 1.

The assessment of the methodological quality of these seven included studies is illustrated in Figure 2.

### 3.2. Characteristics of Included Studies

Seven studies were included in this systematic review [18,19,20,21,22,23,24]. Characteristics of these studies with the corresponding weighted means and SD are outlined in Table 1. All the studies were monocentric cohort studies. Three studies had a prospective design [19,20,24], while the other four represented retrospective analyses. A total of 941 patients were included in these seven studies, of whom 517 were male and 413 female (if indicated). One study did not specify biological sex [22], and one false-negative patient had been omitted from the specification of biological sex [19]. The weighted mean age was 60.7 years (standard deviation, 2 years). One study accounted for 63.1% of the included patients [20].

Two studies exclusively included head and neck melanomas [19,22], and one study reported only on melanomas of the trunk or extremities [24]. Overall, with 438 cases (46.5%), melanomas of the extremities were the most common subgroup regarding primary tumor location.

Five studies that elaborated on this point used a concentration of 2.5 mg/mL ICG for local infiltration at the tumor site. However, the camera system used was different in every study, and in three studies more than one camera system was used [19,21].

### 3.3. Characteristics of SLNB Per Tracing Method

The characteristics of the SLNBs per tracing method with the corresponding weighted means and standard deviation are illustrated in Table 2.

In total, 192 metastatic patients were identified by SLNBs, corresponding to 20% of all 941 patients. One study did not specify by which methods the metastatic patients were identified [19]. Out of the remaining 182 metastatic patients in six studies, 167 (92%) were identified by both methods, nine (5%) solely by Tc, and six (3%) solely by ICG.

A total of 2588 SLNs were sampled, out of which 2223 (86%) were identified by both methods, 292 (11%) solely by Tc, and 73 (3%) solely by ICG. A mean of 2.8 SLNs was sampled per patient. One study did not specify the methods used to identify a metastatic SLN [19]. Of the remaining 2390 SLNs in six studies, 221 metastatic SLNs were identified, corresponding to a weighted mean of 9%. Of these, (91%) were identified by both methods, fourteen (6%) solely by Tc, and five (2%) solely by ICG. One study accounted for 66.7% of metastatic patients, 70.6% of SLNs sampled, and 73.8% of metastatic SLNs [20].

One study did not report the length of follow-up or number of recurrences [24]. The mean or median length of follow-up in the remaining six studies including 861 patients was between 1.2 months [23] and 34.4 months [20], with a weighted mean of 31 months. Fourteen patients suffered recurrences during this time. One study did not provide information on the distinct method of identification of metastatic patients [19], so it was excluded from further calculations of the FNR. Furthermore, as one study presented a mean follow-up period of only 1.2 months, it was excluded from further analyses [23]. Thus, the FNR was calculated on the basis of the data from four studies, as demonstrated in Table 2. These studies showed a higher or equal FNR for ICG as compared to Tc. Based on the results of the four studies including 778 patients, the meta-FNR of ICG was 13% and the meta-FNR of Tc was 10%.

The results of the SLNBs were further stratified according to retrospective and prospective studies and presented as odds ratios with corresponding 95% confidence intervals in the following forest plots.

Figure 3 shows the total number of SLNs identified intraoperatively by Tc or ICG out of all the SLNs sampled. Tc identified a significantly higher number of SLNs in two prospective studies (*p* = 0.001) [20,24] and a higher number of SLNs overall. Heterogeneity between the prospective studies was, however, substantial with I^2^ = 68%. ICG identified a higher number of SLNs in retrospective studies, without a significant difference. Overall, no significant difference could be discerned.

Figure 4 displays the number of metastatic SLNs identified by Tc or ICG out of all the SLNs identified by each method. While Tc numerically identified more metastatic SLN, no significant difference was found in retrospective studies, prospective studies, or overall.

Figure 5 illustrates the number of metastatic SLNs identified by Tc or ICG relative to the total number of metastatic SLNs. Tc identified a significantly higher number of metastatic SLNs in prospective studies (*p* = 0.02); however, this difference was not reproducible in the overall analysis.

Figure 6 exhibits the number of metastatic patients identified by Tc or ICG out of the total number of metastatic patients. No significant difference was found in retrospective studies, prospective studies, or overall.

More metastatic patients, SLNs, and metastatic SLNs were identified by Tc than by ICG. However, the ratio of identification of metastatic patients and metastatic SLN from Tc to ICG was not as pronounced as that of identification of sampled SLNs from Tc to ICG.

Figure 7 illustrates the number of false-negative patients missed by Tc or ICG out of the total number of true-positive and false-negative patients with corresponding risk differences. The pooled risk difference was 0.03, meaning that, based on the available literature, the calculated risk for not identifying a metastatic SLN with ICG compared to Tc would increase by 3%. This would theoretically translate into a number needed to harm in all the patients of 155.6 (see Table 3). However, this difference was not significant.

## 4. Discussion

Both Tc and ICG identified SLNs, metastatic SLNs, and metastatic patients who would have been missed by the other method alone, indicating that no single modality achieves perfect results during SLNBs. SPECT-CT has already been demonstrated to identify a higher number of SLNs than LS [25]. The factors reported to limit SLN identification with Tc include the shine-through effect, a perturbance of SLN identification near the primary tumor by radioactive background signals around the injection site [26]. The area of shine-through has been reported as up to 11cm from the primary tumor in the case of breast cancer [27], which might explain the relatively high FNR of Tc in patients with head and neck melanomas [25].

Tc identified a significantly higher number of SLNs and of metastatic SLNs out of the total number of metastatic SLNs in the prospective studies. This may be due to advantages in study design compared to retrospective studies, such as a more structured approach, methodological standardization, specialized training, and adherence to protocols by medical staff. Yet, results in the prospective group were also majorly influenced by the significant sample size of the study published by Knackstedt et al. in 2021 [20]. Therefore, it also primarily represents the experience of this group, which may differ from other centers.

However, Tc did not identify a significantly higher number of metastatic SLNs out of all SLNs identified by Tc. This indicates that more SLNs are sampled based on Tc, yet this does not directly result in a correspondingly higher number of identified metastatic SLNs or patients compared to ICG.

In the absence of a guideline-compliant radioactive counting rate threshold, surgeons might indeed sample SLNs with a low probability of actually containing metastases [25]. Indeed, different definitions exist to intraoperatively define SLNs on the basis of Tc, including acoustic signal, counts of at least 25 in 10 s, or sentinel to non-sentinel lymph node count ratio [27]. An increasing number of sampled SLNs is associated with an increasing incidence of complications such as seromas [28], wound infections, nerve injuries, or lymphedemas [29]. Therefore, a method that samples a minimal number of SLNs to limit morbidity, while still identifying a maximum of metastatic patients, is desirable. On the other hand, previous publications have also shown that the excision of multiple SLNs might lead to higher rates of positivity and lower FNRs, at least for the subgroup of head and neck melanomas [30,31]. However, this could also simply hint at an insufficient specificity of the current diagnostic standard.

No significant difference between Tc and ICG was found regarding the number of identified metastatic patients. Since the initiation of guideline-compliant adjuvant therapy is determined by the presence of SLN metastasis, independent of the number of metastatic SLNs, the correct identification of metastatic patients rather than the exact number of metastatic SLNs certainly is clinically more relevant when comparing the diagnostic accuracy of these two modalities.

Likewise, no significant difference between Tc and ICG was found regarding the FNR, which, due to the therapeutic consequences mentioned above, is of utmost significance to affected patients. Substantial debate persists on the clinical relevance and consequences of metastatic SLNBs and false-negative SLNBs. Indeed, immediate complete lymph node dissection after a positive SLNB does not increase melanoma-specific survival [32], and no significant difference in overall survival was found between false-negative and true-positive patients [33]. Other authors have shown that false-negative SLNBs lead to significantly lower 2-year overall survival in head and neck melanomas [31], as well as significantly lower overall survival in melanomas of the trunk and extremities compared to true-positive SLNBs [34]. It has to be noted that the potential effects of adjuvant immune checkpoint inhibitors in stage III melanoma, essential to current guidelines [35], have not yet been investigated in these studies. Given the clinical success of these immunotherapeutic options, it seems reasonable to assume that a false-negative SLNB is of significant clinical relevance and that the FNR is indeed an adequate criterion to judge the accuracy of the SLNB. In that regard, ICG may thus offer a safe alternative to Tc for intraoperative guidance in SLNBs in cutaneous melanoma patients.

An increased FNR after a Tc-based SLNB has been associated with several factors. These include technical factors such as the above-mentioned shine-through effect [26] and the lack of implementation of reverse transcriptase-PCR for SLN examination, which has been shown to identify a higher number of positive SLNs as compared to conventional histopathological methods [25]. Interestingly, to the best of our knowledge, none of the studies examined in this systematic review carried out a review of the histopathological results using this method after the diagnosis of a false-negative patient. Other factors are patient- or melanoma-related, such as old age at diagnosis or histologic ulceration [34,36], and head and neck locations of primary tumors [25]. Interestingly, reported FNRs of a combined Tc and ICG approach in studies including high proportions of head and neck melanomas were lower than expected [37], highlighting the role of ICG in this subgroup.

On the other hand, there are also relevant obstacles to the introduction of ICG in medical practice. As, currently, it will most likely not directly replace Tc-based LS, the acquisition of the NIRFI machine and training of the surgical staff is a significant additional financial investment. There is also a considerable learning curve to using the system and reliably performing oncologic surgery under ICG guidance. Additionally, regulatory approval requirements for ICG for this indication may create further barriers.

Limiting factors for identifying SLNs with ICG have also been reported and include a high body mass index [38] and an axillary lymph node field [39]. These factors may both be due to the limited transcutaneous (“before skin incision”) visibility of ICG, which offers a penetration depth reported at 1 cm [40], 1.5 cm [24], or 2 cm [41] only. Furthermore, most of the current studies localize SLNs with Tc-based LS and confirm these pre-localized basins transcutaneously with ICG or confirm SLNs only intraoperatively without blinding the surgeon to the LS results until after the excision of all SLNs identified by ICG. Only one study included in our systematic review reported transcutaneous detection rates [24]. While such a combined, complementary approach of using Tc and ICG represents the clinical standard in many centers, it may also lead to an overestimation of the diagnostic accuracy of ICG. If the operation is carried out using both modalities simultaneously, the fact that an SLN displays ICG positivity after surgical removal does not automatically mean that it would have been found in a true ICG-only approach. Transcutaneous detection rates of SLNs with ICG alone have been reported at 21% [24], 63.4% [39], or 79.4% [38], with a detection rate approaching 100% intraoperatively (“after skin incision”) [24,39]. If transcutaneous identification rates could be further increased, not only would identification rates of SLNs and patients be optimized, but ICG would enable surgeons to perform a more targeted incision and dissection of lymphatic tissue and thus minimize morbidity. Indeed, previous authors have needed fewer drains postoperatively when using ICG [42]. While the data available in the current literature only allowed us to quantitatively compare the intraoperative detection rates of the two methods, transcutaneous identification of the SLN, which is equally important to the whole process, will be crucial in deciding whether ICG-only ever becomes a viable alternative to the current Tc standard.

The seven studies in this systematic review show some limitations that may serve as a guide for future study approaches. A lack of multicenter studies or studies with a larger cohort size constitutes the most important limitation. A substantial proportion of our total cohort is attributable to one study only [20]. Furthermore, to the best of our knowledge, there is no randomized controlled trial to compare the distinct FNRs of Tc and ICG to date. Standardization of the approach is lacking with multiple camera systems across studies, and sometimes there was no indication of the ICG dose used. Finally, some studies did not define the FNR used nor include adequate follow-up time to detect potential recurrences. Large, prospective, randomized controlled trials, adequately separating the two modalities during the procedure, are needed to truly differentiate the diagnostic accuracies of ICG and Tc.

## 5. Conclusions

No significant differences between ICG and Tc were found regarding the most important outcome measures, i.e., the identification of metastatic patients and FNRs. Both methods present characteristics limiting their ability to perform an optimal SLNB. However, ICG may be a non-inferior alternative to Tc for intraoperative guidance in SLNBs in cutaneous melanoma patients in the context of an advantageous adverse events profile, cost considerations, and limited availability of Tc and LS equipment in parts of the world. Our results will need to be confirmed in future randomized controlled trials.

## Figures and Tables

**Figure 1 cancers-16-02523-f001:**
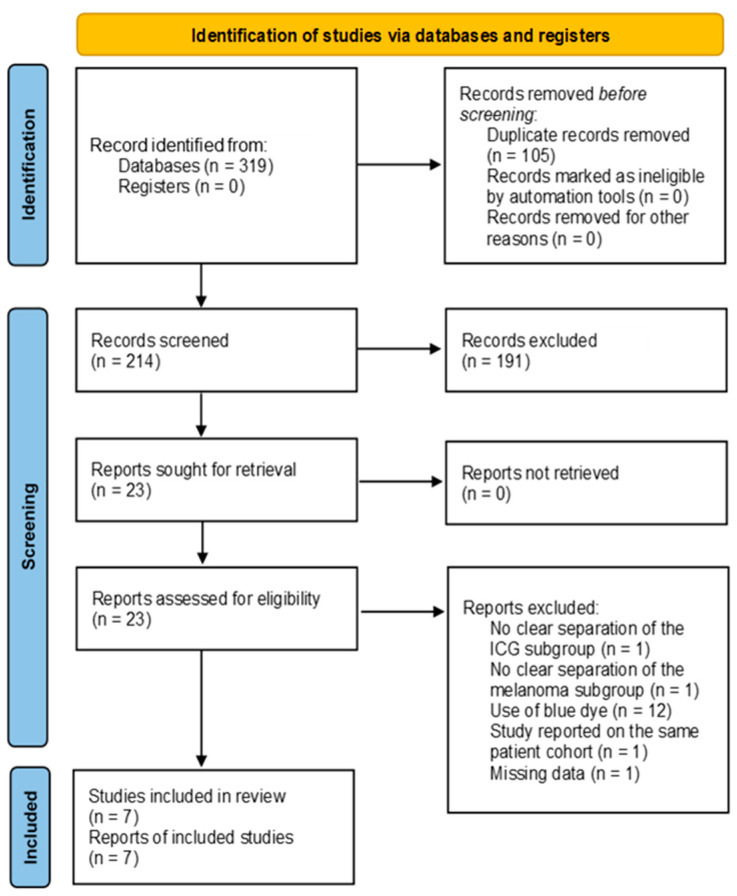
PRISMA 2020 flow diagram of the database search [10].

**Figure 2 cancers-16-02523-f002:**
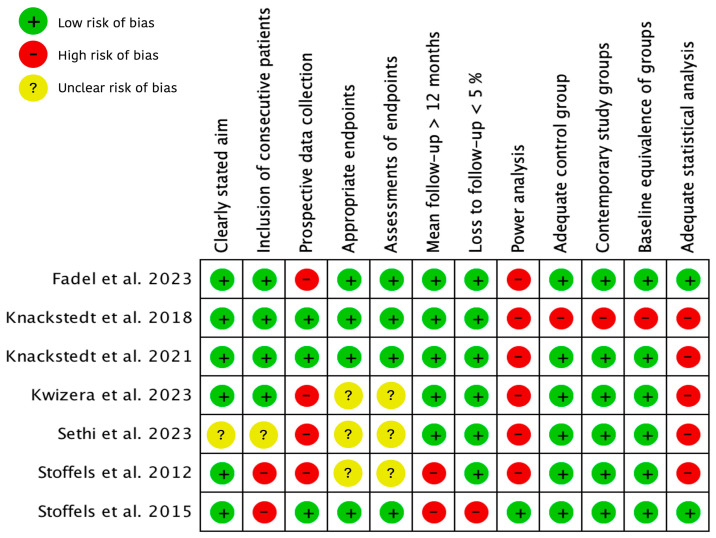
Methodological quality of included studies using the Methodological Index for Non-Randomized Studies (MINORS). + = 2 points, adequately/total agreement; ? = 1 point, reported but inadequate/partial agreement; − = 0 points, not reported/no agreement [14,18,19,20,21,22,23,24].

**Figure 3 cancers-16-02523-f003:**
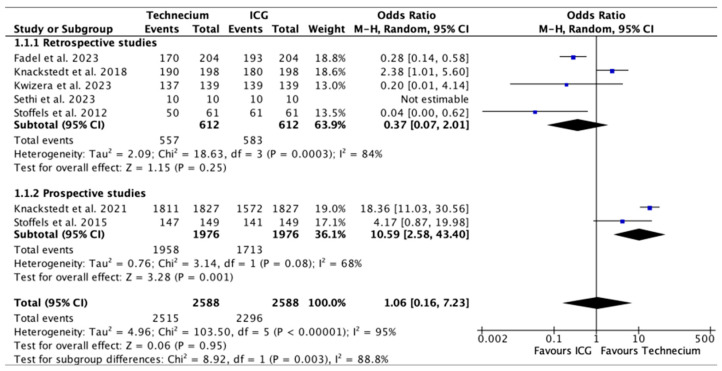
Forest plot depicting effect estimates regarding the total number of SLNs identified by Tc or ICG during SLNBs out of all SLNs sampled, stratified according to retrospective and prospective studies by odds ratio. Blue square: point estimate of the effect for a single study sized according to study weight, black line: confidence interval, diamond: subgroup or overall effect estimate [18,19,20,21,22,23,24].

**Figure 4 cancers-16-02523-f004:**
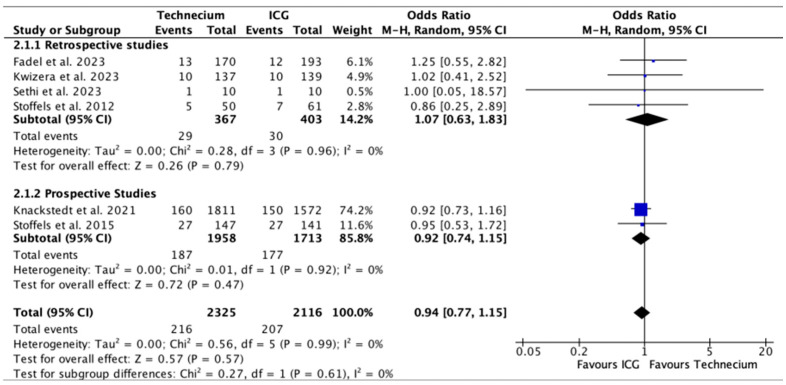
Forest plot depicting effect estimates regarding the number of metastatic SLNs identified by Tc or ICG out of all the SLNs identified by each method, stratified according to retrospective and prospective studies by odds ratio. Blue square: point estimate of the effect for a single study sized according to study weight, black line: confidence interval, diamond: subgroup or overall effect estimate [18,20,21,22,23,24].

**Figure 5 cancers-16-02523-f005:**
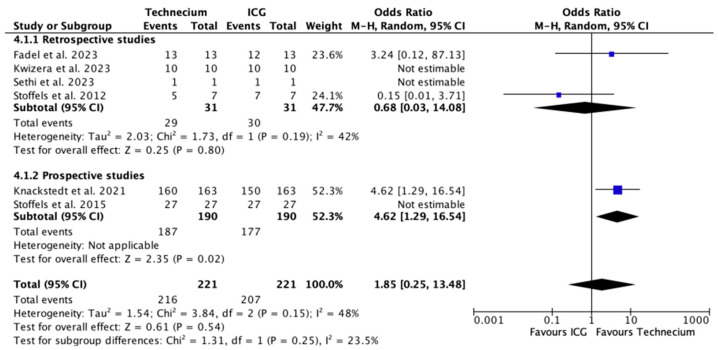
Forest plot depicting effect estimates regarding the number of metastatic SLNs identified by Tc or ICG relative to the total number of metastatic SLNs, stratified according to retrospective and prospective studies with odds ratio. Blue square: point estimate of the effect for a single study sized according to study weight, black line: confidence interval, diamond: subgroup or overall effect estimate [18,20,21,22,23,24].

**Figure 6 cancers-16-02523-f006:**
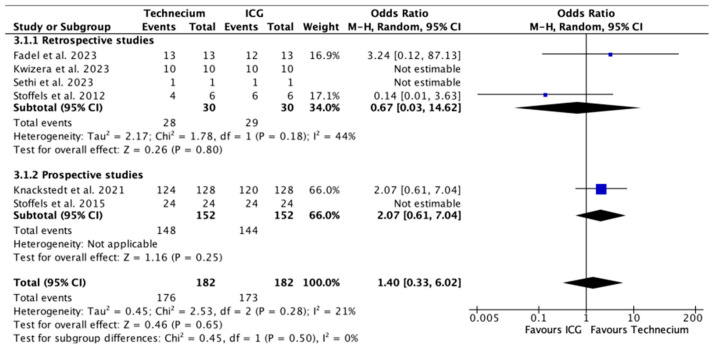
Forest plot depicting effect estimates regarding the number of metastatic patients identified by Tc or ICG out of the total number of metastatic patients, stratified according to retrospective and prospective studies by odds ratio. Blue square: point estimate of the effect for a single study sized according to study weight, black line: confidence interval, diamond: subgroup or overall effect estimate [18,20,21,22,23,24].

**Figure 7 cancers-16-02523-f007:**
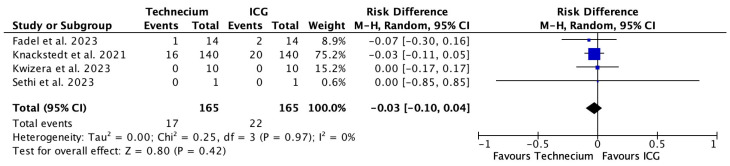
Forest plot depicting effect estimates regarding the number of false-negative patients missed by Tc or ICG out of the total number of true-positive and false-negative patients with corresponding risk differences. Blue square: point estimate of the effect for a single study sized according to study weight, black line: confidence interval, diamond: overall effect estimate [18,20,21,22].

**Table 1 cancers-16-02523-t001:** Characteristics of included studies.

	Kwizera 2023 [21]	Fadel 2023 [18]	Sethi 2023 [22]	Stoffels 2015 [24]	Stoffels 2012 [23]	Knackstedt 2021 [20]	Knackstedt 2018 [19]	Total (%) or Weighted Mean (SD)
Type	Cohort Study	Cohort Study	Cohort Study	Cohort Study	Cohort Study	Cohort Study	Cohort Study	
Design	Retrospective	Retrospective	Retrospective	Prospective	Retrospective	Prospective	Prospective	
Location	Monocentric	Monocentric	Monocentric	Monocentric	Monocentric	Monocentric	Monocentric	
Study period (months)	NR	51	46	18	NR	72	36	
No. patients	52	122	10	80	22	594	61	941 (100)
Men (%)	26 (50)	47 (38.5)	NR	52 (65)	10 (45.5)	337 (56.7)	45 (73.8)	517 (55.6)
Women (%)	26 (50)	75 (61.5)	NR	28 (35)	12 (54.5)	257(43.3)	15 (24.6)	413 (44.4)
Mean age (years)	63	60.5	65	55.5	51.6	61.2	64.3	61 (2)
Primary tumor location								
Head and Neck (%)	11 (21.2)	13 (10.7)	10 (100)	0 (0)	2 (9.1)	136 (22.9)	61 (100)	233 (24.8)
Trunk (%)	20 (38.5)	38 (31.1)	0 (0)	40 (50)	9 (40.9)	163 (27.4)	0 (0)	270 (28.7)
Extremities (%)	21 (40.4)	71 (58.2)	0 (0)	40 (50)	11 (50)	295 (49.7)	0 (0)	438 (46.5)
Camera system	Stryker Elite, SPY-PHI	SPY-PHI	SPY Elite	Fluobeam	PDE	SPY, PDE, Quest	SPY, PDE	
ICG dose (mg/mL)	2.5	2.5	2.5	NR	NR	2.5	2.5	

NR: not reported. SD: standard deviation.

**Table 2 cancers-16-02523-t002:** Characteristics of SLNBs per tracing method.

	Kwizera 2023 [21]	Fadel 2023 [18]	Sethi 2023 [22]	Stoffels 2015 [24]	Stoffels 2012 [23]	Knackstedt 2021 [20]	Knackstedt 2018 [19]	Total (%) or Weighted Mean (SD)
No. SLNs sampled (%)	139	204	10	149	61	1827	198	2588 (100)
identified by both methods	137 (98.6)	159 (77.9)	10 (100)	139 (93.3)	50 (82)	1556 (85.2)	172 (86.9)	2223 (85.9)
identified only by Tc	0 (0)	11 (5.4)	0 (0)	8 (5.4)	0 (0)	255 (14)	18 (9.1)	292 (11.3)
identified only by ICG	2 (1.4)	34 (16.7)	0 (0)	2 (1.3)	11 (18)	16 (0.9)	8 (4)	73 (2.8)
Follow-up (months)								
Mean	NR	NR	16.2	NR	1.2	34.4	30.6	31 (7)
Median	24	24	NR	NR	NR	NR	NR	
No. metastatic patients (%)	10	13	1	24	6	128	10	192 (100)
identified by both methods	10 (100)	12 (92.3)	1 (100)	24 (100)	4 (66.7)	116 (90.6)	NR	167 (91.8)
identified only by Tc	0 (0)	1 (7.7)	0 (0)	0 (0)	0 (0)	8 (6.3)	NR	9 (4.9)
identified only by ICG	0 (0)	0 (0)	0 (0)	0 (0)	2 (33.3)	4 (3.1)	NR	6 (3.3)
No. metastatic SLNs (%)	10	13	1	27	7	163	NR	221 (100)
identified by both methods	10 (100)	12 (92.3)	1 (100)	27 (100)	5 (71.4)	147 (90.2)	NR	202 (91.4)
identified only by Tc	0 (0)	1 (7.7)	0 (0)	0 (0)	0 (0)	13 (8)	NR	14 (6.3)
identified only by ICG	0 (0)	0 (0)	0 (0)	0 (0)	2 (28.6)	3 (1.8)	NR	5 (2.3)
No. recurrences ^(1)^	0	1	0	NR	0	12	1	14 (100)
FNR ICG (%)	0	14	0	NR	NR	14	NR	13 (4)
FNR Tc (%)	0	7	0	NR	NR	11	NR	10 (3)

NR: not reported. SD: standard deviation. ^(1)^: No. patients with recurrences in previously sampled, negative SLN basins.

**Table 3 cancers-16-02523-t003:** FNRs for all patients and FNRs for metastatic patients of ICG and Tc with corresponding relative risk (RR) and number needed to harm (NNH).

FNR	ICG	Tc	RR (95% CI)	NNH (95% CI)	*p* Value
Pooled FNR for all patients (n = 778)	2.8%	2.2%	1.3 (0.7, 2.4)	155.6 (45.5, ∞)	0.419
Pooled FNR for metastatic patients (n = 165)	13.3%	10.3%	1.3 (0.7, 2.3)	33 (10.0, ∞)	0.396

## Data Availability

For data supporting the reported results, please contact the corresponding author.

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
