# Peer review of "Sentinel Lymph Node Detection in Cutaneous Melanoma Using Indocyanine Green-Based Near-Infrared Fluorescence Imaging: A Systematic Review and Meta-Analysis"

_cancers, 2024, doi:10.3390/cancers16142523_

Round 1
Reviewer 1 Report
Comments and Suggestions for Authors The main question addressed by the research is if ICG can be use as unique technique for SLND. Auhtors have exsposed the current evidence and conclusions are accurate. Compared with other published material, the paper add the quality of meta-analysis to subject area. Conclusions are consistent with the results of the forrest-plots. A good performed meta-analysis, it is perfect.Author Response
The main question addressed by the research is if ICG can be use as unique technique for SLND. Auhtors have exsposed the current evidence and conclusions are accurate. Compared with other published material, the paper add the quality of meta-analysis to subject area. Conclusions are consistent with the results of the forrest-plots. A good performed meta-analysis, it is perfect.
- Thank you very much for your review and the positive evaluation.
Reviewer 2 Report
Comments and Suggestions for Authors
Dear Authors,
I have reviewed your manuscript titled "Positive sentinel lymph node (SLN) biopsy of cutaneous melanoma patients" and would like to provide feedback on its quality and suggest areas for improvement.
- A recent metaanalysis Yin R, Ding L, Wei Q, Zhou Y, Tang G and Zhu X: Comparisons of ICG‑fluorescence with conventional tracers in sentinel lymph node biopsy for patients with early‑stage breast cancer: A meta‑analysis. Oncol Lett 21: 114, 2021) showed that ICG is a superior tracer compared to blue dye (BD) or radio-isotope (RI) alone and is not inferior to the combination of BD and RI.
The study supports the widespread clinical use of ICG for SLNB due to its improved detection rates and lower false negative rates.
Given these results, it is expected to be widely adopted in clinical practice for SLNB in breast cancer due to its advantages in tracer performance, safety profile, and operational efficiency. However, these advantages have not been as clearly demonstrated in melanoma. Authors can explore if this difference could be due to differences in tumor biology, study focus, technical application, clinical practice variations, and regulatory processes. Further research specifically targeting melanoma is necessary to confirm the benefits of ICG in this context.
- Could you specify the challenges related to ICG availability in medical centers? Initial costs, such as acquiring NIR equipment and training staff, can be expensive. Not all centers have the necessary technology to detect and use ICG. Additionally, regulatory approval for ICG use can be a lengthy process. The lack of experience with ICG among medical staff can also be a significant barrier. Surgeons can use ICG during surgeries to identify sentinel lymph nodes, while radiologists can play an important role in preoperative diagnosis and planning
- Could the authors explain why Tc identified a significantly higher number of SLNs and metastatic SLNs only in prospective studies? Possible reasons might include the structured design and control in prospective studies, technological and methodological advancements, specialized training and adherence to protocols by medical staff, comprehensive patient follow-up and monitoring, and differences in data quality and completeness between prospective and retrospective studies
Comments on the Quality of English Language
Strengths:
1. Clarity and Precision: The manuscript excels in clarity and precision, especially in the summary and abstract sections. The ideas are well-structured, presenting a logical and coherent flow.
2. Methodological Rigor: Your methodology is well-articulated, adhering to PRISMA and MOOSE guidelines.
3. Comprehensive Review: The systematic review and meta-analysis are thorough, considering a substantial number of initial studies and applying stringent inclusion and exclusion criteria.
4. Quantitative Presentation: The results are effectively presented with quantitative data, utilizing tables and forest plots that facilitate understanding and comparison.
Areas for Improvement:
1. Writing and Style:
Simplify some sentences to improve the text’s fluency and readability.
Eliminate repetitions, especially between the summary and the abstract.
Reduce the excessive use of acronyms; ensure all acronyms are defined the first time they are used.
2. Depth in Discussion:
The discussion section could benefit from a more in-depth exploration of the study’s limitations, including clinical and technical factors, in addition to the design and sample size.
Provide more detailed suggestions on how future research can address these limitations.
3. Structure and Format:
Review the structure of some sections to improve coherence and avoid repetition. For instance, some methodological and result aspects could be summarized or reorganized for greater clarity.
Best regards,
Author Response
I have reviewed your manuscript titled "Positive sentinel lymph node (SLN) biopsy of cutaneous melanoma patients" and would like to provide feedback on its quality and suggest areas for improvement. A recent metaanalysis Yin R, Ding L, Wei Q, Zhou Y, Tang G and Zhu X: Comparisons of ICG‑fluorescence with conventional tracers in sentinel lymph node biopsy for patients with early‑stage breast cancer: A meta‑analysis. Oncol Lett 21: 114, 2021) showed that ICG is a superior tracer compared to blue dye (BD) or radio-isotope (RI) alone and is not inferior to the combination of BD and RI. The study supports the widespread clinical use of ICG for SLNB due to its improved detection rates and lower false negative rates. Given these results, it is expected to be widely adopted in clinical practice for SLNB in breast cancer due to its advantages in tracer performance, safety profile, and operational efficiency. However, these advantages have not been as clearly demonstrated in melanoma. Authors can explore if this difference could be due to differences in tumor biology, study focus, technical application, clinical practice variations, and regulatory processes. Further research specifically targeting melanoma is necessary to confirm the benefits of ICG in this context.
- Thank you for adding this relevant reference regarding the evidence in breast surgery. It has been added and the respective paragraph was updated accordingly. The success of ICG for SLNB in breast cancer has certainly been a major motivating factor for us to conduct this study and to investigate the potential benefits of ICG in other fields of oncologic surgery. However, we feel that an in-depth discussion of why the results may vary between breast cancer and melanoma patients would go beyond the scope of this publication, especially because this would heavily rely on speculation as solid evidence is sparse. Further research specifically targeting melanoma is certainly necessary.
- Could you specify the challenges related to ICG availability in medical centers? Initial costs, such as acquiring NIR equipment and training staff, can be expensive. Not all centers have the necessary technology to detect and use ICG. Additionally, regulatory approval for ICG use can be a lengthy process. The lack of experience with ICG among medical staff can also be a significant barrier. Surgeons can use ICG during surgeries to identify sentinel lymph nodes, while radiologists can play an important role in preoperative diagnosis and planning
- Thank you for this input. The discussion section was adapted to include these considerations: “On the other hand, there are also relevant obstacles to the introduction of ICG in medical practice. As currently it will most likely not directly replace Tc-based LS, the acquisition of the NIRFI-machine and training of the surgical staff is a significant additional financial investment. There is also a considerable learning curve to using the system and reliably performing oncologic surgery under ICG-guidance. Additionally, regulatory approval requirements for ICG for this indication may create further barriers.” (s. l. 367 – 372).
- Could the authors explain why Tc identified a significantly higher number of SLNs and metastatic SLNs only in prospective studies? Possible reasons might include the structured design and control in prospective studies, technological and methodological advancements, specialized training and adherence to protocols by medical staff, comprehensive patient follow-up and monitoring, and differences in data quality and completeness between prospective and retrospective studies
- The discussion section was adapted to elaborate on these considerations: “This may be due to advantages in study design compared to retrospective studies such as a more structured approach, methodological standardization, specialized training and adherence to protocols by medical staff. Yet, results in the prospective group were also majorly influenced by the significant sample size of the study published by Knackstedt et al. in 2021 [23]. Therefore, it also primarily represents the experience of this group which may differ from other centers.” (s. l. 311 – 317).
Strengths:
- Clarity and Precision: The manuscript excels in clarity and precision, especially in the summary and abstract sections. The ideas are well-structured, presenting a logical and coherent flow.
- Methodological Rigor: Your methodology is well-articulated, adhering to PRISMA and MOOSE guidelines.
- Comprehensive Review: The systematic review and meta-analysis are thorough, considering a substantial number of initial studies and applying stringent inclusion and exclusion criteria.
- Quantitative Presentation: The results are effectively presented with quantitative data, utilizing tables and forest plots that facilitate understanding and comparison.
- Thank you very much for taking the time to review our paper, the added value of your inputs and the positive feedback.
Areas for Improvement:
- Writing and Style:
Simplify some sentences to improve the text’s fluency and readability.
Eliminate repetitions, especially between the summary and the abstract.
Reduce the excessive use of acronyms; ensure all acronyms are defined the first time they are used.
- We certainly agree that there is significant repetition between the simple summary and the abstract. However, this format was demanded by the journal and is also new to us. We feel that it is rather difficult to avoid repetition here, but if the editorial office provides us with more guidance on this, we could adapt as requested.
- As suggested, we tried to reduce the number of acronyms where they were not necessary and especially made sure that all of them are defined the first time they are used, both in the abstract and main text.
- Depth in Discussion:
The discussion section could benefit from a more in-depth exploration of the study’s limitations, including clinical and technical factors, in addition to the design and sample size.
Provide more detailed suggestions on how future research can address these limitations.
- We tried to incorporate the limitations of both methods as well as of the study as a whole throughout the discussion. We felt that this structure serves readability better than strictly organizing it into specific sections. Limitations are discussed in several paragraphs as follows:
- Limitations of Tc: 341-349, 358-368, 396-406
- Limitations of ICG: 341-343, 407-417, 417-437
- Limitations of the study as a whole: 351-357, 417-426, 438-448
- The wordcount of the discussion section currently stands at 1358 words. We feel that this is already quite lengthy and would not want to make it too long-winded for the reader. However, if this remains a major concern we can certainly expand on these aspects.
- Structure and Format:
Review the structure of some sections to improve coherence and avoid repetition. For instance, some methodological and result aspects could be summarized or reorganized for greater clarity.
- Thank you for the suggestion. We agree that the methods and results sections are rather lengthy and may appear repetitive in certain parts. However, given the extensive guidelines we had to follow during the preparation of this manuscript (PRISMA, MOOSE, Cochrane Handbook, MINORS), this seemed necessary to fulfill all requirements and present the necessary details regarding the exact process that was followed.
Reviewer 3 Report
Comments and Suggestions for Authors
Woelffer M, et al. conducted a systemic review on the use of Tc vs. ICG in SLNB in melanoma. Although this topic is of interest, the major concern is that both retrospective and prospective cohort studies were included, and the conculsions varied with vs. without retrospective cohort studies. Prospective randomized trials are considered to be the gold standard to establish a new method for diagnosis as they are by design free from biases, if performed strictly and correctly. Restrospective cohort studies, although there are many ways during analyses to decrease potential biases, are still subjected to unobserved biases. So, it is often the case that prospective studies are given more "credibility" when it comes to clinical decision making. With said, as Tc identified significantly higher number of SLN in prospectie studies, I would refrain myself from taking all restrospective studies and revise the conclusion to "ICG may be a non-inferior choice", as it may just be due to observed and/or unobserved biases.
Author Response
Woelffer M, et al. conducted a systemic review on the use of Tc vs. ICG in SLNB in melanoma. Although this topic is of interest, the major concern is that both retrospective and prospective cohort studies were included, and the conculsions varied with vs. without retrospective cohort studies. Prospective randomized trials are considered to be the gold standard to establish a new method for diagnosis as they are by design free from biases, if performed strictly and correctly. Restrospective cohort studies, although there are many ways during analyses to decrease potential biases, are still subjected to unobserved biases. So, it is often the case that prospective studies are given more "credibility" when it comes to clinical decision making. With said, as Tc identified significantly higher number of SLN in prospectie studies, I would refrain myself from taking all restrospective studies and revise the conclusion to "ICG may be a non-inferior choice", as it may just be due to observed and/or unobserved biases.
- Thank you for your review and the helpful comments. We certainly agree that in general a prospective randomized controlled trial provides superior scientific evidence. However, no such study investigating the diagnostic accuracy of ICG versus Tc-based lymphoscintigraphy in cutaneous melanoma patients has been published to date. Given the perioperative practicalities during sentinel-lymph-node-dissection in cutaneous melanoma patients, all relevant studies were non-randomized cohort studies (prospective or retrospective), in which each patient acted as their own control as both methods are used simultaneously during the operation. As these cohort studies were the best scientific evidence available in literature, we had to rely on them for this meta-analysis.
- Furthermore, it must be considered that none of the included studies, regardless of retrospective or prospective study design, were free of potential bias as illustrated by the analysis according to the MINORS-criteria (Table 1).
- Nevertheless, acknowledging that there may be a difference between retrospective and prospective design studies, we performed specific sub-analyses and reported that in our forest plots to provide the reader with a more nuanced analysis of the available data.
- It is true that Tc identified a higher number of SLNs in prospective studies compared with retrospective ones. However, as we tried to elaborate on in the discussion of the results, identifying a high number of lymph nodes may not necessarily be a solely positive aspect. SLN-biopsy is primarily a diagnostic procedure which aims to identify the presence of lymphatic metastasis. Therefore, the number of metastatic patients identified and ultimately the FNR are of superior importance. Here, no difference between prospective and retrospective studies was observed. Achieving the same results by excising more lymph nodes may also be associated with a higher morbidity.
- Additionally, including only prospective studies would heavily skew the results towards a single study (Knackstedt 2021), due to its significant sample size.
- Due to these considerations, we judge it worthwhile to include both retrospective and prospective studies as it provides the reader with a more complete picture of the available evidence.
- The conclusion has been rephrased to “non-inferior alternative” as suggested.
Round 2
Reviewer 3 Report
Comments and Suggestions for Authors
I agree that prospective vs. retrospective cohorts are different. In this regard, the conclusions should be made with caution. One way to do this, is to compare the cohort characteristics between prospective vs. retrospective, and analyze where the discrepancies may reside from, to what specific patient population one of these two analyzed methods can be better adopted. With said, I am not satisfied with how my concerns has been addressed.